# Thermostability profiling of MHC-bound peptides: a new dimension in immunopeptidomics and aid for immunotherapy design

Emma C. Jappe[1,2], Christian Garde[1], Sri H. Ramarathinam [3], Ethan Passantino[3], Patricia T. Illing [3], Nicole A. Mifsud [3], Thomas Trolle[1], Jens V. Kringelum [1✉], Nathan P. Croft [3✉] & Anthony W. Purcell [3✉]

The features of peptide antigens that contribute to their immunogenicity are not well understood. Although the stability of peptide-MHC (pMHC) is known to be important, current assays assess this interaction only for peptides in isolation and not in the context of natural antigen processing and presentation. Here, we present a method that provides a comprehensive and unbiased measure of pMHC stability for thousands of individual ligands detected simultaneously by mass spectrometry (MS). The method allows rapid assessment of intra-allelic and inter-allelic differences in pMHC stability and reveals profiles of stability that are broader than previously appreciated. The additional dimensionality of the data facilitated the training of a model which improves the prediction of peptide immunogenicity, specifically of cancer neoepitopes. This assay can be applied to any cells bearing MHC or MHC-like molecules, offering insight into not only the endogenous immunopeptidome, but also that of neoepitopes and pathogen-derived sequences.

[1] Evaxion Biotech, Bredgade 34E, 1260 Copenhagen, Denmark. [2] Department of Health Technology, Technical University of Denmark, 2800 Lyngby, Denmark. [3] Department of Biochemistry and Molecular Biology, Infection and Immunity Program, Biomedicine Discovery Institute, Monash University, Clayton, VIC, Australia. ✉email: jkgm@evaxion-biotech.com; nathan.croft@monash.edu; anthony.purcell@monash.edu

CD8[+] T cell recognition of epitopes relies upon target cells processing protein antigens into peptides and presenting these on the cell surface in complex with major histocompatibility complex (MHC) molecules [human leukocyte antigen (HLA) in humans][1,2]. Despite the multitude of potential peptides in a given protein that may theoretically bind MHC, only a fraction of these may actually be presented as a complex on the cell surface[3,4]. Moreover, of these naturally presented peptides, even fewer will be capable of eliciting a T cell response[5]. In the context of patient-specific T cell immunotherapy in cancer, identifying not only the peptides that will be presented on the surface of the tumor but also the most efficacious targets—the immunogenic neoepitopes—remains a major challenge[6–8]. The use of MS to sequence and identify naturally processed and presented peptides (immunopeptidomics) has provided large qualitative, and in a limited number of cases, quantitative datasets[9]. However, these studies are yet to describe definitive features of pMHC presentation that can predict immunogenicity[10–12]. Indeed, current prediction algorithms only take a selection of peptide features into account, and assays for the identification of features linked to peptide immunogenicity typically study them in isolation[13,14]. The stability of pMHC has been linked to immunogenicity in several studies[6,8,14]. However, despite this feature impacting on the composition of the immunopeptidome, it is difficult to extract this information for individual peptides since their presence is dictated by features of peptide generation, source antigen abundance and turnover, MHC-binding characteristics, and complex stability.

Inspired by the work of Nordlund and colleagues[15,16] who probed the thermostability of whole proteomes, here we develop a method to generate thermostability curves across entire immunopeptidomes. The method relies upon modification of established immunopeptidomics workflows and rapid thermal treatment of samples prior to utilizing an optimized immunoprecipitation assay for thermostable native peptide HLA complex (pHLA) isolation, peptide elution, and quantitative data-independent acquisition-mass spectrometry (DIA-MS). As such, we provide evidence of highly robust thermostability data on two monoallelic cell lines, achieving a distribution of stability curves for >1,000 peptides per allele. We find that the obtained measure of thermostability yields important insights into peptide immunogenicity by training artificial neural network (ANN) models that improve the prediction of immunogenic peptides, specifically cancer neoepitopes.

## Results

**An MS-based assay for stability profiling of the immunopeptidome.** The MS immunopeptidomics workflow we recently described in detail[9] has been optimized to obtain large peptide datasets representing "snapshots" of the peptide repertoire presented by the cell at a given point in time by isolating the pHLA expressed by the cells. We reasoned that we could extend upon this workflow by studying the thermal stability of these complexes and that these modified conditions would result in temperature-dependent recovery of specific pHLA, allowing a stability measure for individual peptide ligands to be determined. Based on these considerations, we developed a pHLA stability assay that applies a modified microscale immunopeptidomics workflow and DIA-MS approach to generate thermal stability curves for naturally processed and presented immunopeptidomes (Fig. 1).

To develop this assay, we used the HLA class I low-expressing C1R cell line[17,18] modified to express high levels of either HLA-A*02:01 or HLA-B*07:02 (Supplementary Fig. 1). Despite the low surface expression of endogenous HLA I (HLA-B*35:03 and HLA-C*04:01) by parental C1R cells, there is no impairment in their antigen processing and presentation capacity making transfected cells essentially monoallelic, antigen-presenting cells[9]. HLA-A*02:01 and HLA-B*07:02 were selected as these represent common HLA allotypes[19].

Prior to carrying out the workflow pertaining to the stability profiling of the immunopeptidome, we constructed spectral libraries from immunopeptidomics data generated using the C1R-A*02:01 and C1R-B*07:02 cell lines and more conventional data acquisition strategies[9]. This enabled post-acquisition peptide spectrum matching of DIA-MS data obtained for the stability treated samples. Spectral libraries of more than 8,000 peptides per allele were generated based on immunoaffinity purification of pHLA complexes, isolation of their peptide cargo, and sequencing of these eluted peptides by high-resolution data-dependent acquisition (DDA)-based MS using published workflows[9]. Peptide identity was established using PEAKS Studio 8.5®[20] processing (Fig. 1a,b, Supplementary Data 1 and Supplementary Data 2).

For stability profiling of the immunopeptidome, we developed a microscale variation of the optimized immunopeptidomics approach described by Purcell et al.[9] The microscale workflow was carried out by lysing C1R cells expressing either HLA-A*02:01 or HLA-B*07:02, clearing lysates and separating these into aliquots of $5 \times 10^7$ cell equivalents (Fig. 1c). Aliquots were incubated for 10 min in triplicate at different temperatures in the range 37–73 °C. We selected this temperature range and the incubation time empirically. Inspired by results from previous work[6,21–23], we designed preliminary experiments to determine the incubation time that would result in complete ablation of peptide signal, indicative of complete pHLA dissociation at high incubation temperature, yet enable sufficient peptide coverage at 37 °C (Supplementary Fig. 2). We tested two different incubation times, 5 min and 10 min, at temperature points 37 °C, 60 °C, 70 °C, and 80 °C. An incubation time of 10 min revealed a defined temperature endpoint could be achieved at 70 °C whilst retaining satisfactory peptide recovery at 37 °C (Supplementary Fig. 2).

Next, the effect of heating of C1R-A*02:01 or C1R-B*07:02 cell lysates across the selected temperature range of 37–73 °C was investigated by isolating the pHLA complexes using the pan-HLA I antibody W6/32 after each thermal treatment and analyzing the eluted peptides in DIA mode. Samples were analyzed using a DIA strategy with fixed isolation window size of 24 m/z (Fig. 1d). HLA-specific spectral libraries were built in Skyline and used to match DIA data obtained from the thermally treated samples (Fig. 1e). DIA-MS data were filtered in Skyline to include only peptide sequences of 8–11 amino acids in length as these constitute the majority of MHCI-associated ligands[2]. The fold-change in peak area for individual peptides as a result of increasing temperature was determined based on the peak area at the selected reference temperature (37 °C). MS chromatographic peak areas were normalized based on indexed retention time (iRT) internal standard peptides spiked into samples.

Upon inspection of the normalized DIA-MS data for the thermally treated samples, we observed a sigmoidal decay trend (Fig. 1f and Supplementary Fig. 3), and the normalized data were therefore fitted to a logistic sigmoid function (for details, see Methods). Despite the stringent filtering criteria selected, this yielded >1,000 peptide-specific sigmoidal melt curves for both HLA-A*02:01 and HLA-B*07:02 allotypes.

**The kinetic stability of pHLA is closely linked to thermostability.** Several studies have linked kinetic pMHC stability to immunogenicity[8,14,24], and a strong correlation between thermal and kinetic stability of pHLA has recently been demonstrated using differential scanning fluorimetry (DSF)[6]. To justify the use of a thermostability measure to describe the stability of the pHLA

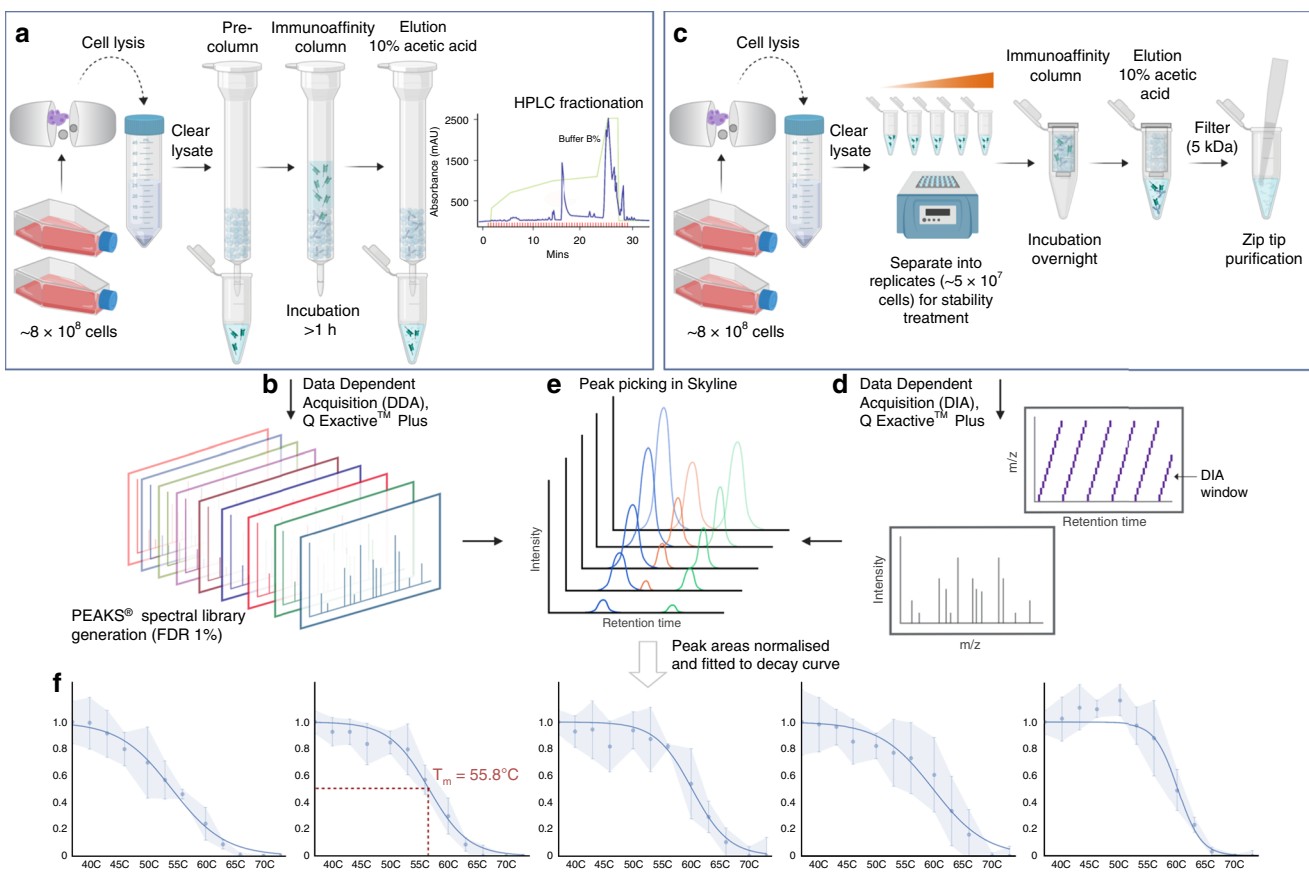

**Fig. 1 Workflow for stability profiling of HLA-associated peptides. a** Initially, immunoprecipitation on C1R cells expressing the HLA allele of interest was carried out by culturing cells, lysing them, clearing the lysate, and isolating peptides according to established workflows[9]. **b** Liquid chromatography-tandem mass spectrometry (LC-MS/MS) analysis of pHLA eluates was performed in DDA mode to create HLA allele-specific spectral libraries. **c** Small-scale immunoprecipitation was carried out by clearing lysates and separating these into replicates consisting of $5 \times 10^7$ cells, after which aliquots were incubated in triplicate at temperatures ranging from 37 to 73 °C with a temperature step-size of 3-4 °C. **d** Subsequently, the remaining thermostable HLA-bound peptides were eluted, filtered, and analyzed using a DIA strategy to enable peptide quantification at different temperature points. **e** Spectral library matching and filtering were performed in Skyline. **f** Peptide peak areas for triplicate samples were normalized and fitted to sigmoidal decay curves to determine the temperature at which half of the complex was unfolded, termed the thermal melting temperature ($T_m$). Thermal treatment of C1R cell lysates was carried out at 12 different temperature points ranging from 37 to 73 °C with $n=3$ biological replicates at each temperature point. Data are presented as median values ± SD. FDR: False Discovery Rate. Parts of the figure were generated using BioRender.com.

complex, we attempted to replicate these findings and applied the microscale immunoprecipitation approach illustrated in Fig. 1 to study the kinetic stability of pHLA complexes eluted from C1R-A*02:01 cells in a time rather than temperature-dependent manner (Supplementary Note 1). These time-dependent samples were analyzed in DIA-MS mode and peptide spectra were matched to the HLA-A*02:01-specific spectral library using Skyline. Assuming that all complexes are intact at the initial time point (0 hrs), this point was used as reference to calculate and compare the fold-change in peak signal after different incubation times for individual peptides. Peak areas were normalized and fitted to exponential decay curves to calculate peptide half-lives ($t_{1/2}$). We found a good correlation between $t_{1/2}$ and $T_m$ in our study (Spearman correlation coefficient = 0.79), supporting previous findings[6] and demonstrating that thermostability is a surrogate for kinetic stability (Supplementary Fig. 4).

**Extracting a thermostability measure from pHLA-specific melt curves.** We verified that the length of the peptides for which sigmoidal melt curves could be constructed based on the DIA data followed a typical length-distribution for both alleles (Fig. 2a)[11,18,23]. Thus, no bias in peptide length was introduced in

the thermostability measurements. From these data, the stability of each pHLA complex was inferred by calculating its thermal melting temperature ($T_m$) – the temperature at which 50% of the complex is unfolded (Supplementary Data 3 and Supplementary Data 4)[21]. We found no correlation between the attained measure of thermostability and the median peak area at 37 °C, demonstrating that the results were not merely an artifact of the ionization efficiency of the peptide[25] (Supplementary Fig. 5). Furthermore, $T_m$ values for peptides restricted by HLA-A*02:01 and HLA-B*07:02 showed that the thermostability for these alleles is not generally length-dependent (Fig. 2b).

**Revealing inter-allelic and intra-allelic differences in thermostability.** Although considered monoallelic, the C1R cell lines used in these experiments also express low levels of HLA-C*04:01[18]. Whilst the expression of HLA-C*04:01 is typically considered to hamper the investigation of introduced HLA alleles[18], in this study the presence of HLA-C*04:01 was leveraged to assess assay robustness (Fig. 2c and Supplementary Fig. 3) and for a comparison of the distribution of $T_m$ values across all three HLA loci (Fig. 2d). For the robustness analysis, we considered the correlation between $T_m$ values for the HLA-C*04:01

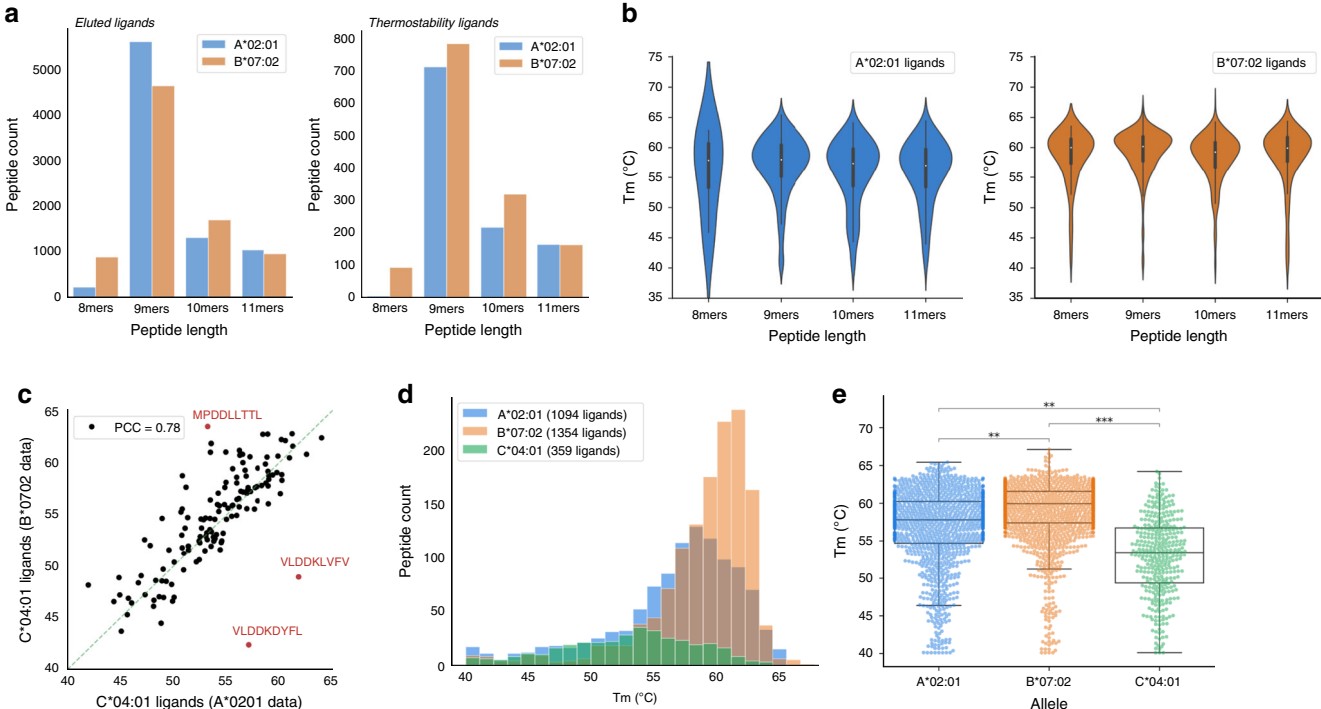

**Fig. 2 Significant inter-allelic differences are observed in $T_m$ distributions.** The length distributions of peptides bound to either HLA-A*02:01 or HLA-B*07:02 identified in the thermostability assay and the immunoprecipitation of pHLA complexes using established workflows[9] were as expected from previous immunopeptidomics studies, with ligands being predominantly 9-mers. **b** $T_m$ values for 1094 HLA-A*02:01-restricted peptides do not differ significantly based on their length ($p = 0.016$, Kruskal-Wallis test; individual length comparisons with Mann-Whitney U test with Bonferroni adjustment for multiple comparisons, $p > 0.05$). $T_m$ values for 1354 HLA-B*07:02-restricted peptides do not differ significantly for all but one of the length comparisons, with a difference observed between the $T_m$ distributions of 9mers and 10mers; $p = 5.20*10^{-5}$, Kruskal-Wallis test; individual length comparisons with Mann-Whitney U test with Bonferroni adjustment for multiple comparisons, $p > 0.05$ for all but the 9mer and 10mer distributions with $p = 1.20*10^{-6}$). Violin plot representation of data shows the median as center white dot and 25th percentile and 75th percentile as bounds of the center black box. **c** There is a good correlation between the $T_m$ values for HLA-C*04:01-restricted peptides identified in either the C1R-A*02:01 or C1R-B*07:02 assay (PCC = 0.78) demonstrating high assay robustness. This correlation is further improved (PCC = 0.87) when removing outlier peptides (indicated in red) that have high binding affinity to other alleles expressed by the C1R cell line. **d** $T_m$ distributions for HLA-A*02:01 (blue), HLA-B*07:02 (orange) and HLA-C*04:01 (green), expressed by the C1R cell lines studied show that there is a substantial difference in their variance (transparency is given to the bars to visualize their overlap and allele-specific distributions). **e** A significant difference in the $T_m$ distribution for peptides restricted by HLA-A*02:01 (1094 peptides), HLA-B*07:02 (1354 peptides) and HLA-C*04:01 (359 peptides) is observed; $p = 1.48*10^{-93}$, Kruskal-Wallis test; **$p < 10^{-25}$, ***$p < 10^{-50}$, Mann–Whitney U test with Bonferroni adjustment for multiple comparisons. Box plot representation of data shows the median as center, 25th percentile and 75th percentile as bounds of boxes, maximum as 75th percentile + 1.5 times the interquartile range and minimum as 25th percentile - 1.5 times the interquartile range. PCC: Pearson Correlation Coefficient.

peptides identified in both the C1R-A*02:01 and C1R-B*07:02 assays and found a strong correlation (Pearson Correlation Coefficient = 0.78) (Fig. 2c and Supplementary Data 5). Intriguingly, we observed that the outlier peptides in the two assays had high predicted binding affinity to one of the other 'competing' alleles expressed by the cell lines (Fig. 2c and Supplementary Table 1), offering unique insights into the potential competition that occurs between alleles expressed by a given cell line for available peptide ligands. $T_m$ values for individual ligands across the three alleles varied from 40.1 °C to 67.1 °C, with a median of 57.8 °C, 59.9 °C and 53.5 °C for HLA-A*02:01, HLA-B*07:02 and HLA-C*04:01, respectively (Fig. 2d). A comparison of the distribution of $T_m$ values of all peptides across all three allotypes revealed that the stability of naturally presented peptide ligands varies significantly inter-allelically (Fig. 2e). HLA-C*04:01-bound peptides had the lowest average $T_m$, consistent with a number of prior biochemical studies[26,27], as well as reports demonstrating lower cell surface expression levels and greater ER-retention of HLA-C alleles[26,28]. Moreover, we observed that intra-allelic $T_m$ values varied in their level of dispersion, with HLA-C*04:01 showing the highest variance (Fig. 2d,e).

**Thermostability profiling provides added data dimensionality.** To assess whether intra-allelic variance in pHLA $T_m$ could be explained by ligand affinity, we predicted peptide binding affinities using NetMHCpan-4.0[11] and correlated these with thermostability measurements. This analysis showed a poor correlation (Fig. 3a), with the majority of the eluted ligands predicted to have high binding affinity to their cognate HLA allele. Our ability to discriminate these peptides using the thermostability assay, therefore, provides an additional dimension of information (Fig. 3a). This led us to explore whether we could tease apart sequence features that drive peptide stability. For this, we trained ANN models based on transformed thermostability data (Fig. 3b), which enabled the identification of binding motifs in the larger eluted ligand datasets (Fig. 3c). We observed a distinction between the motifs of the high and low stability binders when predicting eluted ligands with our stability model, which could not be identified when predicting the ligands using NetMHCpan-4.0 (Fig. 3c). The information content in the peptide-binding motifs was higher for the more stable binders compared to the less stable binders for both HLA-A*02:01 and

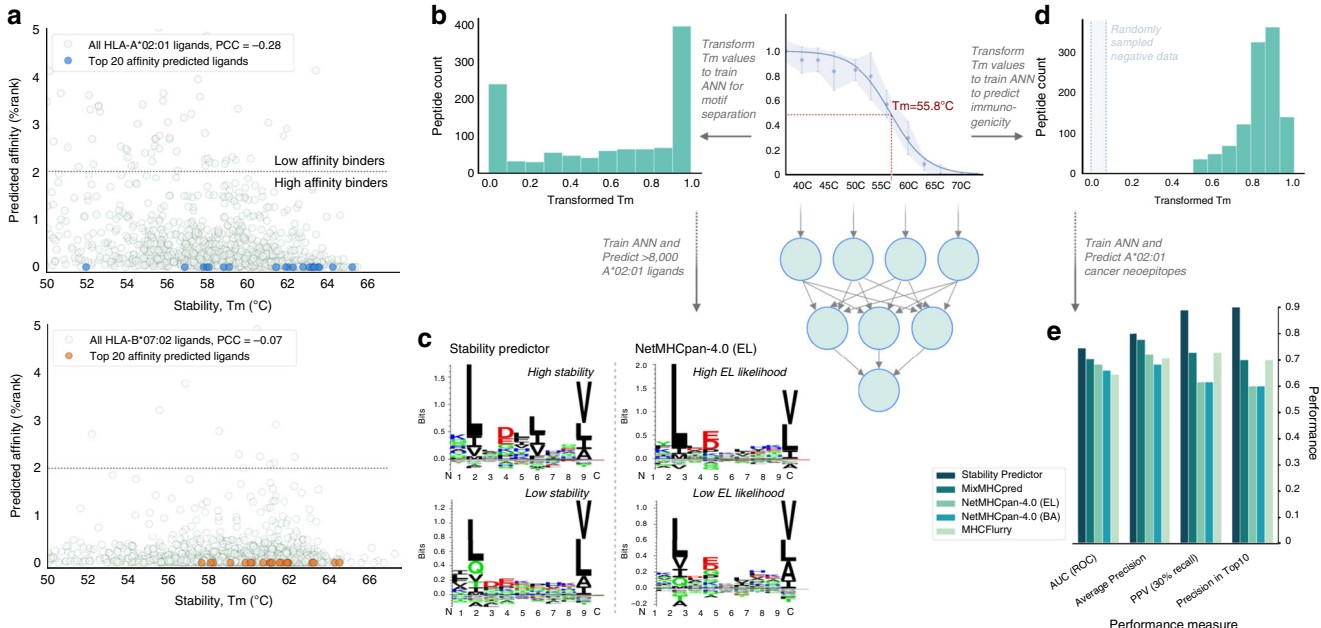

**Fig. 3 Thermostability data of HLA-ligands improves immunogenicity prediction. a** The information content of the stability data for each of the two main alleles studied (HLA-A*02:01 and HLA-B*07:02) was investigated by predicting peptide binding affinity with netMHCpan-4.0 (BA)[11] of the 8-11mer eluted ligands for which stability measurements were achieved demonstrating no correlation between predicted affinity and measured stability. Thus, granularity through peptide stability for predicted high-affinity HLA ligands was observed for both alleles. **b** $T_m$ values were calculated based on thermal melt curves for the identified peptides across 12 different temperature points ranging from 37 °C to 73 °C with $n$=3 biological replicates at each temperature point. Thermal melt curve data are presented as median values ± SD. We trained an ANN model using transformed $T_m$ values as input for the identified 1094 peptides restricted by HLA-A*02:01. The resulting models were used to predict >8,000 allele-specific eluted ligands. Binding motifs were constructed using the 1500 best and poorest predictions in both models. **c** This demonstrated a separation in the motif identified for high and low stability binders which could not be achieved using netMHCpan-4.0 (EL). **d** We investigated whether the additional dimensionality of the thermostability data could improve the prediction of immunogenic cancer neoepitopes by training ANNs with $T_m$ values rescaled to the interval [0.5;1] as positive training data and length-balanced, randomly sampled peptides from the human UniProt-Swissprot as negative training data. **e** The resulting Stability Predictor demonstrates superior performance to current prediction tools with 9 of the predicted peptides in the top 10 being true neoepitopes (Precision in Top10). BA: binding affinity. PCC: Pearson Correlation Coefficient. EL: eluted ligand. AUC: Area Under the Curve. ROC: Receiver Operating Characteristic. PPV: Positive Predictive Value.

HLA-B*07:02 (Fig. 3c and Supplementary Fig. 6). While anchor positions (P2 and P9) were similar between peptides predicted to have high and low stability, the difference in binding motifs for HLA-A*02:01 peptides was striking at P4 and P6 – a difference not observed when predicting the ligand likelihood using NetMHCpan-4.0. Interestingly, the largest difference between immunogenic and non-immunogenic pMHCs has previously been demonstrated to be at these central positions[29], which have been shown to be in close contact with the T cell receptor and important for T cell recognition[30,31]. These findings collectively highlight the limitations of current binding affinity and eluted ligand likelihood prediction algorithms.

**Thermostability data improve the prediction of cancer neoepitopes.** We investigated whether the stability data encompassed information that could directly improve the prediction of T cell epitopes with the focus in this work being specifically on cancer neoepitopes. We used the HLA-A*02:01 data to train an ANN model, since the discrepancy between the motifs of high and low stability HLA-A*02:01 binders was more prominent and indicative of high information content in the stability data, and sufficient HLA-A*02:01-restricted neoepitope data were available to evaluate the trained model. Thus, an HLA-A*02:01 Stability Predictor was trained using transformed $T_m$ values as positive data and randomly sampled, length-balanced peptides from the human UniProt-Swissprot database as negative data

(Supplementary Data 3 and Fig. 3d). We tested this model on a dataset of 26 cancer neoepitopes curated from the literature by Blaha et al.[6] and 20 cancer peptides confirmed to be negative in multiple subjects tested in multimer/tetramer or ELISPOT assays, retrieved from the Immune Epitope Database (IEDB)[32] (Supplementary Data 6). This negative dataset is considered to consist of "difficult negatives" as they were predominantly investigated based on being anticipated HLA binders, thus making it challenging for a prediction model to distinguish the positive and negative datasets. We established that our Stability Predictor is superior to the state-of-the-art prediction tools[33], NetMHCpan-4.0[11], MixMHCpred[34], and MHCFlurry[12], in distinguishing immunogenic neoepitopes from non-immunogenic cancer peptides across all performance measures with nine of the predicted top 10 peptides being true immunogenic neoepitopes (Fig. 3e and Supplementary Fig. 7). Achieving such high precision in neoepitope prediction remains crucial for the optimal design of personalized T cell immunotherapies in cancer. Of note, the Stability Predictor was trained using significantly less data than the vast amount of binding affinity and eluted ligand data used to train the prediction tools included in this benchmark[11,12,34]. Due to the limited size of the negative test dataset, we carried out an additional benchmark analysis to ensure robustness in our results in which we retrieved all confident negatives (199 peptides) from the IEDB, including all cancer, autoimmune and viral peptides. Here, we show that the Stability Predictor significantly outperforms

current prediction algorithms for all model comparisons (AUC $p < 0.05$). This leads us to hypothesize that the same trend will be evident when predicting a larger neoepitope dataset.

## Discussion

The work herein represents an important step towards expanding our current understanding of peptide immunogenicity. We have developed a method to obtain quantitative stability data on naturally processed and presented MHC-associated peptide ligands using a modified immunopeptidomics workflow and targeted MS approach. The ease of implementation and use of the method makes it highly accessible for the field of immuno-peptidomics. Combined with a tailored bioinformatics pipeline, the method enables the generation of thermostability curves for endogenous pMHC ligands in a simultaneous and unbiased manner. By extracting pMHC-specific thermostability measures for >1000 peptide sequences per allele, we have shown that the method provides added dimensionality to the data we typically derive from "snapshot" immunopeptidomics studies. We demonstrate both intra- and inter-allelic variance in stability profiles which may hold the potential to discriminate competition for binding of promiscuous peptides. Importantly, we show that by incorporating thermostability data for naturally presented ligands we improve the prediction of immunogenic cancer neoepitopes. These findings are of great relevance as we are currently challenged in identifying the most efficacious targets from a list of predicted high-affinity MHC ligands.

Our findings are supported by previous reports suggesting that pMHC stability is a promising feature for neoepitope prioritization[6,8]. Multiple studies have demonstrated a correlation between pMHC stability and peptide immunogenicity and, in some instances, even shown that pMHC stability is a better predictor of immunogenicity than pMHC affinity[14,24], which has been attributed to the importance of prolonged exposure of the complex to circulating T cells[24,35].

To date, most studies utilizing pMHC stability as a feature to better guide the prediction of peptide immunogenicity have focused on the kinetic stability of the complex[14,24,36,37]. As demonstrated previously and shown in this work, another means of studying the stability of a ligand-protein interaction is through changes in the thermostability of the protein as a result of ligand binding[38], which has been leveraged to probe the thermostability of whole proteomes[15,16]. Multiple studies have investigated the thermostability specifically of MHC molecules with different peptides bound within the binding groove; however, it has not previously been possible to study the thermodynamics of extensive, naturally processed and presented peptide repertoires, and the majority of studies looking into pMHC thermostability have not investigated this as a direct measure of immunogenicity[6,21–23]. In addition to this, assays for pMHC stability analysis rely on the ability to re-fold MHC heavy chain and β2m in vitro and require pre-selection and synthesis of peptides. The latter is a major downside of current affinity and stability assays[13,14,21], as the selection of peptides is typically based on prior knowledge of peptide affinity profiles. The method described in this work eliminates this bias as the natural processing of pMHC has been allowed to proceed prior to the assay.

Although we here focus on endogenous pHLA repertoires from cultured cell lines, the versatility and ease of use of the method makes it applicable to all types of cells expressing MHC from any species, provided an antibody exists to immunoprecipitate the complex for analysis. This, therefore, allows the investigation of any MHC molecule in any context and can be readily extended to investigate peptide presentation in cancer, autoimmunity, or infectious disease. Although here we used a DIA-MS method, the

approach can be adapted to more sensitive assays such as multiple reaction monitoring, which is ideally suited to detecting low copy-number peptides. Thus, the method would enable the study of the stability of peptide repertoires presented by cancerous cells, and how these are affected by varying levels of IFNγ exposure[39,40], or the stability of repertoires presented by virally infected cells, and how such repertoires change during an infectious cycle[41]. Particularly, mouse models for virus infection are ideal for studying features of pMHC and T cell immunogenicity because they are so well established and highly tractable[3,41]. In addition to this, the method could be applied to study the effect of post-translational modifications on the stability of pMHC binding, which is currently an unexplored area of research.

In the future, we foresee the assay having a clear application in generating stability measurements for neoepitopes from patient-derived cell lines or biopsies to drive a better selection of immunotherapeutic targets. In addition, measuring the extent to which the stability of pathogen-derived pMHC correlates with known CD8+ T cell responses will only serve to bolster our fundamental understanding of peptide immunogenicity.

## Methods

**Cell lines and culture.** The class I-reduced B-lymphoblastoid C1R cell line (ATCC CRL-1993) has reduced expression of endogenous HLA-A*02:01 and HLA-B*35:03 and normal expression of HLA-C*04:01[17,18] and was used for the generation of monoallelic cell lines expressing either HLA-A*02:01 or HLA-B*07:02. C1R-A*02:01 is a transfectant cell line, generated as described in[42], and C1R-B*07:02 is retrovirally transduced using established transduction methodologies[43]. Cell lines were cultured in RPMI 1640 media (Thermo Fisher Scientific, Waltham, MA) supplemented with 10% heat-inactivated fetal calf serum (Sigma-Aldrich, USA), 1 mM MEM sodium pyruvate, 2 mM L-glutamine, 100 mM MEM non-essential amino acids, 5 mM HEPES buffer solution, 55 mM 2-mercaptoethanol, 100 U ml$^{-1}$ penicillin and 100 mg ml$^{-1}$ streptomycin; purchased from Gibco (Thermo Fisher Scientific), at 37 °C, 5% CO2. In addition, C1R-A*02:01 transfectants were maintained under hygromycin (0.3 mg ml$^{-1}$) selection during cell culture. Cells were tested for mycoplasma contamination, and continued HLA class I expression was confirmed using flow cytometry after staining with W6/32 (pan HLA class I-specific monoclonal antibody produced in-house from W6/32 hybridoma, ATCC HB-95), and Goat F(ab')2 Anti-Mouse IgG(H + L), Human ads-PE (1:500, catalog number 1032-09, Southern Biotech, USA). Once cells had grown to high density, they were harvested in batches of $4 \times 10^8$ cells by centrifugation ($520 \times g$, 10 min, 4 °C) and washing in ice-cold phosphate-buffered saline (PBS), after which the pellets were snap-frozen in liquid nitrogen and stored at -80 °C until further use.

**Purification of pHLA complexes to generate spectral library.** Peptide spectral libraries of HLA-A*02:01 and HLA-B*07:02 were generated based on the isolation of pHLA complexes and subsequent dissociation of bound peptides using the immunoprecipitation protocol described in detail in[9], using $8 \times 10^8$ cells. Briefly, cells were lysed by homogenization followed by detergent-based lysis and incubation with rotation for 45 min at 4 °C. The lysate was centrifuged for 10 min at 2,000 × g, 4 °C, after which the supernatant was transferred to a pre-chilled ultra-centrifuge tube and centrifuged for 45 min (100,000 × g, 4 °C). The pHLA complexes were immunoaffinity purified from the cell lysate supernatant using either the HLA-A*02:01-specific antibody BB7.2 (ATCC HB-82, grown and purified in-house) or the pan-HLA I antibody W6/32 (ATCC HB-95, grown and purified in-house) crosslinked to protein A sepharose (antibody to protein ratio of 10 mg ml$^{-1}$) as described in[9]. Bound complexes were eluted with 5 ml 10% acetic acid, and the eluted peptides, class I heavy chain and β2-microglobulin (β2m) were fractionated on a 4.6 mm internal diameter × 100 mm long monolithic reversed-phase (RP) C18 high-performance liquid chromatography (HPLC) column (Chromolith SpeedROD, Merck Millipore, Germany) on an ÄKTAmicro™ HPLC system (GE Healthcare, UK; Unicorn v5.11 software). After loading samples under mobile phase conditions of 98% buffer A (0.1% v/v trifluoroacetic acid (TFA) in water) and 2% buffer B (80% v/v acetonitrile (ACN), 0.1% v/v TFA in water), peptides were enriched using a gradient of buffer A to B running at 1 ml min$^{-1}$ with gradient conditions of 2–40% B over 4 mins, 40–45% B over 4 min and 45–99% B over 2 min, and collected in 500 µl fractions. Fractions were pooled into nine peptide-containing pools which were concentrated by vacuum centrifugation and reconstituted in 2% v/v ACN, 0.1% v/v formic acid (FA) in water. To carry out retention time prediction in down-stream DIA-MS analyses, 200 fmoles of iRT peptides were spiked into each fraction pool[44]. Pooled fractions were sonicated for 10 mins, centrifuged for 10 m at 21,000 × g, and stored at -80 °C until LC-MS/MS data acquisition.

**Microscale immunoprecipitation for pHLA stability analysis.** Peptide thermal stability was analyzed using a microscale immunoprecipitation protocol modified from the workflow described previously[9]. Pellets of $4 \times 10^8$ C1R-A*02:01 or C1R-B*07:02 cells were lysed by cryogenic milling and subsequent resuspension of homogenized cell material in 10 ml lysis buffer as described above. Cell lysates were incubated for 45 min at 4 °C with slow end-over-end mixing after which lysates were cleared by centrifugation at $3700 \times g$ for 10 min at 4 °C. Cleared lysates were separated into replicates consisting of $5 \times 10^7$ cell equivalents in LoBind Eppendorf tubes, which were then centrifuged for 10 min at $21,000 \times g$ (4 °C) to ensure complete clearing of each replicate lysate. The cleared lysates were transferred to new Eppendorf tubes and incubated for 10 min in triplicate at different temperatures (37 °C, 40 °C, 43 °C, 46 °C, 50 °C, 53 °C, 56 °C, 60 °C, 63 °C, 66 °C, 70 °C or 73 °C), using a benchtop heat block (Benchmark Scientific isoBlock™). Upon completion of the thermal incubation, samples were placed immediately on ice. Microscale immunoprecipitation of thermally treated pHLA complexes was then carried out by mixing cooled lysates with W6/32 antibody (400 µg per replicate) bound to protein A sepharose, incubating overnight at 4 °C and then centrifuging through MobiSpin Columns (MoBiTec GmbH, Germany) with inserted filters of 10 µm pore size, with subsequent and extensive washing by addition of PBS. Bound pHLA complexes were eluted with 300 µl 10% acetic acid and the cell eluate, consisting of eluted peptides, class I heavy chain, $\beta_2$m and W6/32 antibody, was filtered using pre-washed (twice with 450 µl 10% acetic acid) 5 kDa centrifugal filter units (Ultrafree®-MC-PLHCC, Merck Millipore, Germany). Filter units were centrifuged at $16,000 \times g$ for 60 min to collect sample flow-through, and filters were washed with an additional 200 µl 10% acetic acid to ensure that all residual peptides had passed through the filter. 200 fmoles iRT peptide mixture was spiked into the samples for downstream retention time prediction and peak normalization. The filtered peptide solution was purified and buffer exchanged prior to LC-MS/MS analysis using ZipTip Pipette tips with a C18 bed inserted into a 100 µl tip (Agilent, OMIX A57003100) and eluted in 30% ACN/0.1% FA. The purified samples were concentrated by vacuum centrifugation and subsequently reconstituted in 2% v/v ACN, 0.1% v/v FA in water, and stored at -80 °C. Prior to LC-MS/MS analysis, samples were thawed, sonicated for 10 min, and centrifuged for 10 min at $21,000 \times g$.

**Data acquisition by LC-MS/MS.** LC-MS/MS analysis of pHLA eluates was performed on a Q-Exactive Plus Hybrid Quadrupole Orbitrap (Thermo Fisher Scientific) coupled to a Dionex UltiMate 3000 RSLCnano system (Thermo Fisher Scientific) with data acquisition for the reconstituted fraction pools from large-scale immunoprecipitations being achieved by DDA-MS, and data acquisition for the microscale immunoprecipitations concerning pHLA stability being analyzed using a DIA strategy[45,46]. Data were acquired using Xcaliber 3.0.63 acquisition software (Thermo Fisher Scientific). For DDA analysis, 6 µl of each concentrated fraction pool was loaded onto a Dionex Acclaim PepMap100 200-mm C18 Nano-Trap Column with 100-µm internal diameter (5-µm particle size, 300-Å pore size) in buffer A (2% v/v ACN, 0.1% v/v FA in water) at a flow rate of 15 µl min$^{-1}$. HLA-B*07:02-associated peptides were separated on a Dionex Acclaim RSLC PepMap RSLC C18 column (50-cm length, 75-µm internal diameter, 2-µm particle size, 100-Å pore size) and subsequently eluted at a flow rate of 250 nl/min over an increasing gradient of buffer B (80% v/v ACN, 0.1% v/v FA in water) of 2.5–7.5% over 3 min, 7.5–37.5% over 120 min, 37.5–42.5% over 3 min, 42.5–99% over 5 min and 99% over 6 min after which the gradient dropped to 2.5% buffer B over 1 min, before re-equilibrating at 2.5% for 20 min. Data were collected in positive mode with an MS1 resolution of 70,000 and scan range 375–1,575 m/z and an MS2 resolution of 17,500 with scan range 200–2,000 m/z. The top 20 ions of charge state 2–5 per cycle were chosen for MS/MS with a dynamic exclusion of 15 s. HLA-A*02:01-associated peptides were eluted with the same flow rate over an increasing gradient of buffer B (80% v/v ACN, 0.1% v/v FA in water) of 2.5–7.5% over 1 min, 7.5–35% over 40 min, 35–99% over 5 min, 99% over 6 min, and then dropping to 2.5% buffer B over 1 min and finally re-equilibrating at 2.5% for 20 min. Data were collected as for HLA-B*07:02-associated peptides; however, with MS1 scan range 375–1,800 m/z and with the top 12 ions per cycle selected for MS/MS.

For DIA analysis, 6 µl of each thermally treated sample replicate was loaded onto the trap column and eluted from the C18 column at a flow rate of 250 nl min$^{-1}$ over the same gradient as above for DDA. The mass spectrometer was operated with an MS1 resolution of 70,000 and scan range 375-1,575 m/z followed by 25 DIA scans with fixed isolation window size of 24 m/z in the range 387.426 to 987.6988 m/z at a resolution of 17,500.

**Spectral library generation in PEAKS Studio®.** PEAKS Studio® (v.10)[20] was used to process the DDA-MS data from nine fraction pools of HLA-eluted peptides resulting from immunoprecipitation of $8 \times 10^8$ C1R cells[9]. DDA data files were imported with Instrument set to Orbitrap, Fragmentation HCD, and no digestion enzyme. Precursor and fragment mass tolerances of 10 ppm and 0.02 Da, respectively, were selected, and the DDA spectra were searched against the human UniprotKB database (v2019-08) with iRT peptide sequences used as contaminant database. Analysis was carried out with oxidation [+15.99] and deamidation [+0.98] set as variable peptide modifications, with a maximum of three modifications per peptide. A false discovery rate (FDR), determined based on a target-

decoy database, of 1% was used to generate the HLA-specific spectral libraries in PEAKS Studio®.

**DIA data analysis and spectral library matching in Skyline.** Skyline v.4.2[47] was used to process the DIA data for all stability treated replicates. Only peptide sequences of 8–11 amino acid residues in length were included[2]. The DDA data from PEAKS Studio® was used to build spectral libraries, and retention time alignment was carried out by recalibrating iRT standard values relative to the peptides being added and selecting a time window of 10 min. The DIA isolation scheme was specified based on isolation windows in the DIA raw files and retention time filtering included only scans within 10 min of the predicted retention time. The raw DIA files were imported into Skyline and processed using the HLA-specific spectral libraries to extract fragment ion peak areas. Due to high complexity of the data, poor peptide transitions were removed. Transitions were removed based on whether or not they were observed in the 37 °C replicates as this is the temperature point at which the maximal number of peptides with the maximal peak areas were expected to be observed. Thus, transitions that did not have a coeluting peak for all 37 °C replicates were removed as well as peptides for which the isotopic dot product (idotP) value for all 37 °C samples was blank.

**Pre-processing of thermostability data.** MS chromatographic peak areas for the filtered peptide datasets were normalized based on iRT internal standard peptides spiked into all samples. Total peak areas $A$ for each peptide were normalized by a factor $f$ defined as the average of the mean-centered iRT peptide peak areas

$$A_{norm} = \frac{A}{f}, \text{ where } f = \frac{1}{J} \sum_j \frac{x_{ij}}{\frac{1}{I} \sum_i x_{ij}} \qquad (1)$$

where $j$ denotes the iRT peptide and $i$ the replicate at any given matrix position. For replicates with dotP < 0.8, peak areas were set to 0. The median peak areas for each time or temperature point in the stability treatment protocol were outlier corrected, with each corrected peak area being the mean of the median peak area at any given time or temperature point and the median of peak areas at adjacent points. The peptide datasets were filtered to remove peptides for which the median dotP of the 37 °C triplicates or the 0 hr triplicates was < 0.8 as well as iRT peptide fragments and in-house contaminant peptides catalogued over many experimental controls.

**Generating thermostability curves.** For the temperature-dependent microscale immunoprecipitation samples, fold-changes in the median value of the normalized, outlier-corrected peak areas resulting from DIA analysis were computed using the lowest temperature point (37 °C) as reference. Non-linear least squares were used to fit logistic sigmoid functions to the peak area fold-changes as a function of temperature, $T$

$$f(T) = \frac{1}{1 + e^{s \cdot (T - T_m)}} \qquad (2)$$

where $T_m$ is the transition midpoint for pHLA complex unfolding. The slope of the curve at the transition midpoint is defined as the first derivative of $f(T)$ for $T = T_m$, which when solved shows that slope = –s/4. The value of $f(T)$ for T = 37 °C was fixed to one for all peptides.

The peptides were filtered to the set with fits satisfying $R^2 > 0.85$. This was satisfied by 86% of the peptides in the C1R-A*02:01 data and 82% of the peptides in the C1R-B*07:02 data. Endogenous ligands expressed naturally by parental C1R cells were identified by intersecting the two datasets. This set was supplemented with ligands in the C1R background dataset, defined below. The GibbsCluster algorithm v2.0[48] was used to cluster data and remove any additional sequences that were clearly outliers in respect to the HLA-A*02:01 and HLA-B*07:02 motifs, respectively. This yielded a total of 1,094 peptides and associated thermal stability curves for HLA-A*02:01 and 1,354 for HLA-B*07:02.

**Filtering eluted ligands contained in the spectral library.** Eluted ligands were filtered for overlapping sequences between the HLA-A*02:01 and HLA-B*07:02 datasets and sequences in the stability data, described above. Furthermore, the eluted ligands were filtered based on known contaminants as well as the established C1R background, defined below. GibbsCluster v2.0[48] was employed to flag and remove spurious ligands. This yielded a total of 8,138 and 8,134 eluted ligands for HLA-A*02:01 and HLA-B*07:02, respectively.

**C1R background and analysis of assay robustness.** All post-processed peptides from the HLA-A*02:01 and HLA-B*07:02 were compiled, and the sequences were clustered[48] to identify motifs characteristic of the HLA-C*04:01 allele, which is expressed at relatively low levels, and the HLA-B*35:03 allele, expressed at residual levels, by C1R cells[18]. Only ligands identified in the C1R background dataset, comprising ligands in the work by Schittenhelm et al.[18] and in-house identified C1R ligands, were included. HLA-B*35:03 peptides were subsequently removed from further analysis, as these represented just 73 peptides. In the comparison of $T_m$ values between the two assays, the likelihood of being an eluted ligand for outlier peptides was predicted using NetMHCpan-4.0[11]. The distribution of $T_m$ values for each of the alleles was compared statistically using the Kruskal-Wallis

test for significance and, as post hoc test, the Mann Whitney test with Bonferroni adjustment of $p$-values to correct for multiple comparisons.

**Data transformation and artificial neural network training**. Analyses to investigate whether the thermostability data encompassed information that could help tease apart sequence features that drive peptide stability (i) and improve the prediction of peptide immunogenicity (ii) were carried out by training ANN model ensembles. The stability ($T_m$) values were transformed in order to be used as input for the ANN models as described below.

(i) Binding motifs of highly and lowly stable binders were identified through ANN training using only the peptide sequences for which stability data was obtained to train the models. $T_m$ value transformation to train the ANN ensembles was carried out such that stability measurements were rescaled to the interval [0;1], ensuring clustering around 0 and 1. First, all values were normalized

$$T_{m\_norm} = f(T_m) = \frac{T_m - \text{minimum}(T_m)}{\text{maximum}(T_m) - \text{minimum}(T_m)} \quad (3)$$

Then, the normalized $T_m$ values, $T_{m\_norm}$, were transformed to lie a distance of 3 times the median $T_m$ value from the median, with values < 0, changed to 0, and values > 1, changed to 1

$$T_{m\_trans} = f(T_{m\_norm}) = T_{m\_norm} + 3 \cdot (T_{m\_norm} - \text{median}(T_{m\_norm))) \quad (4)$$

ANN networks were trained with 60, 80, and 100 hidden neurons for 150 epochs using an adapted NNAlign approach with insertions and deletions[10,49,50]. Data were randomly partitioned into 5 partitions, and ANN ensembles were trained using 5-fold nested cross-validation[50] yielding 20 ANN models for each network architecture. The model for each subset of partitions yielding the best performance based on mean squared error (MSE) on the test set, was included in the final network ensemble. The model was used to predict the stability of >8,000 HLA-specific eluted ligands which were pre-processed and filtered, as described above. Sequence motifs were generated using Seq2Logo-2.1[51].

(ii) ANN ensembles were trained using the peptide sequences for which stability data were obtained and their transformed $T_m$ values as positive input (denoted 'Stability Predictor'). The transformation was carried out using a linear normalization approach.

$$f(T_m) = 0.5 \cdot \frac{T_m - \text{minimum}\left(T_{m_{all}}\right)}{\text{maximum}\left(T_{m_{all}}\right) - \text{minimum}\left(T_{m_{all}}\right)} + 0.5 \quad (5)$$

A negative complement to the positive training data was randomly sampled from the human Uniprot-Swissprot database (v2019-04) and assigned a target value of 0. Peptide sampling was carried out in a length-balanced manner, i.e. for each length $k$, $10 \times n$ peptides were sampled, where $n$ indicates the number of ligands of length $k$. We trained ANN ensembles using the adapted NNAlign approach described in (i). Network ensembles were trained with 40, 60, and 80 hidden neurons, respectively, and for 200 epochs. Peptide data were partitioned into five subsets using a clustering approach modified from[52] to minimize the similarity between training and test data. As above, training using 5-fold nested cross-validation yielded 20 ANN models for each network architecture, and the final network ensemble consisted of models with the lowest MSE. The final Stability Predictor constituted ensembles of 60 trained networks each. The predictor was evaluated using a positive dataset of cancer neoepitopes curated from the literature by Blaha et al.[6] which were given the target value 1 and a negative dataset consisting of cancer peptides confirmed to be negative in ELISPOT or multimer/ tetramer assays with >10 subjects tested, retrieved from the IEDB (2019-12). This yielded 26 positive immunogenic neoepitopes and 20 non-immunogenic cancer peptides for HLA-A*02:01. The performance measures used to evaluate the Stability Predictor were AUC (ROC), average precision (AP), positive predictive value (PPV) at 30% recall, and precision in top 10. Model performance was compared to NetMHCpan v4.0[11], MixMHCpred v2.0.2[34], and MHCFlurry v2.0[12]. To compare immunogenic and non-immunogenic peptides, a two-sided, independent samples $t$ test was used.

**Reporting summary**. Further information on research design is available in the Nature Research Reporting Summary linked to this article.

## Data availability

Mass spectrometry proteomics data, PEAKS Studio® search results, and Skyline Report files have been deposited in ProteomeXchange Consortium via the PRIDE[53] partner repository under accession code PXD017824 (C1R-A*02:01 and C1R-B*07:02 DDA LC-MS/MS; https://www.ebi.ac.uk/pride/archive/projects/PXD017824) and PXD017839 (C1R-A*02:01 and C1R-B*07:02 DIA LC-MS/MS for thermal stability experiments and the experiments used to determine complete ablation of peptide recovery at high temperature; https://www.ebi.ac.uk/pride/archive/projects/PXD017839). All other data are available in the article and supplementary information files or from the corresponding authors upon reasonable request. Source data are provided with this paper.

## Code availability

Code for training the HLA-A*02:01 thermostability ANN model was developed using NNAlign[10,49,50]. For re-training of the stability predictor described in this work, we refer to the NNAlign webserver http://www.cbs.dtu.dk/services/NNAlign-2.0/. All training settings are described in the Methods, and the data used to train the predictor are available as Supplementary Data. Any additional code is available from the corresponding authors upon reasonable request.

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

## Acknowledgements

We acknowledge Rochelle Ayala Perez for laboratory assistance in growing and purifying antibodies from hybridoma cell lines, Ritchlynn Aranha for technical assistance with experiments, Shutao Mei for generously sharing data on known HLA-C*04:01 peptides from parental C1R cells and Pouya Faridi for helpful discussions of results. We acknowledge the Monash Proteomics & Metabolomics Facility for the provision of mass spectrometry instrumentation, training, and technical support, as well as the Monash University Flowcore for flow cytometry instrumentation and assistance. We would also like to thank the bioinformatics team at Evaxion Biotech for valuable input and discussions on training the prediction models. Cell culturing, microscale immunoprecipitation, and mass spectrometry related work were conducted at Monash University. The majority of the subsequent data pre-processing and development of machine learning training models were carried out at Evaxion Biotech, Copenhagen. This research was supported by the Innovation Fund Denmark under the Industrial PhD Program (grant 5189-00133B), Monash University, and Evaxion Biotech.

## Author contributions

E.C.J., J.V.K., A.W.P., and N.P.C. conceived the experimental ideas. E.C.J., N.P.C., and P.T.I. designed the experiments with valuable input from S.R., N.A.M. and A.W.P. E.C.J., E.P., P.T.I., N.A.M., and N.P.C. performed experiments and MS analyses, and MS data pre-processing was carried out by E.C.J. under supervision by N.P.C. C.G. and E.C.J. wrote the code and developed the algorithms under supervision by T.T. and J.V.K. E.C.J. made figures and analyzed data with input from N.P.C., C.G., T.T., and A.W.P. E.C.J., A.W.P., N.P.C., and C.G. wrote the manuscript. All authors revised and approved the final manuscript.

## Competing interests

E.C.J., C.G., J.V.K., and T.T. are employed by Evaxion Biotech that holds IP for identifying neoepitopes. A.W.P is on the scientific advisory board and N.P.C. is a specialist advisor at Evaxion Biotech. The remaining authors declare no competing interests.
