## [Peer Review File · Nature Communications]

REVIEWER COMMENTS

Reviewer #1 (Remarks to the Author):

Jappe et al. examine the temperature stability of a global set of peptide ligands interrogated by mass spectrometry. This is an important proxy for the stability of peptide class I binding, which is related to peptide immunogenicity, critically important for improving cancer immunotherapy. In general, this is a fine paper, basically acceptable as is. A few comments for the authors to ponder.

1. C1R cell line is indicated as "low class I" - it is important to inform the readers that this is due to transcription of class I genes and is not related to down regulation of the antigen processing machinery.
2. Does heating of lysates induce protein aggregation, particularly related to class I complexes? Does recovery of soluble class I fractions significantly drop with temperature, and could this potential bias be related to the bound peptide affinity?
3. Since the temperature treatments are done using whole-cell lysates, there are proteases (presumably inhibited, but still...), chaperones, other peptides (even small peptides which might facilitate exchange), and other factors that could impact thermal stability of particular complexes. That's just part of the experimental caveats, which is fine, but would it not have been possible to immunoprecipitate complexes and then subject them to the temperature gradient. Perhaps explain the rationale for choosing your method over others.
4. In Figure 3A, is there a reason to plot % rank rather than just the affinities themselves? To better visualize the potential correlation (or lack thereof)?
5. The predictive power could be used to provide a proof of principle of whether thermo-stability leads to longer-lasting complexes, using algorithm-generated examples of equal-affinity peptides that are predicted to have low or high stability. (i.e. made-up peptides that were not identified by MS). Pulse peptides and track via fluorescent label or MS over time. Not required, but would provide excellent proof of principle.
6. For future studies, consider more physiological exploration of class I peptide complex duration. This could be done by treating cells with BFA for increasing times.

Reviewer #2 (Remarks to the Author):

The manuscript by Jappe et al examines peptide/HLA complexes by using a novel method that takes advantage of mass spectrometry (MS) to measure, simultaneously, the temperature stability of thousands of peptide complexes. The approach quantitates the amount of specific peptide remaining after incubation of antibody-purified HLA complexes at different temperatures. Previous approaches have shown that the thermal stability of a specific peptide/HLA complex correlates with its potential for immunogenicity. The experimentally determined stability has better predictive ability than current algorithms for binding to HLA. However, these past approaches have required use of recombinant HLA and synthetic peptides to determine thermal stability (e.g. using differential scanning fluorimetry, DSF).

The method by Jappe is very novel, as it not only examines thousands of pep/HLA complexes in a single experiment, but it measures the stability of the complexes isolated directly from tumor cells. This enables one to assess not just stability, but the abundance of a specific peptide/HLA complex. This is especially relevant to neoantigens that could be used in vaccine formulations as targets for T cell therapies. The study also showed that they could compare relative stabilities of peptides bound to different HLA alleles. Finally, it is significant that they could use their data to develop improved algorithms for predicting pMHC stability.

The only major comment that I have is that I would like to have the authors address the issues of identifying neoantigens among the eluted pep/HLA complexes, and quantitating their stabilities. While they validate their procedure with some neoantigen/HLA complexes that were previously examined by DSF, the question really is what fraction of these would be of sufficient abundance to analyze with their approach. Actual data from a sequenced tumor cell line should be used to address this, but in absence of this they should at the least discuss this issue of abundance of potential neoantigens in the context of their approach. Minor question, what are the brown bars in Fig 2d?

Reviewer #3 (Remarks to the Author):

This study analyzes thermostability of pMHC as a potential new proxy for immunogenicity. The ideas and suggestions in this study are novel and intriguing and the ms is very well written.

I was asked to focus my comments on the computational aspects of the ms.

In the last part of the study the authors present to predictive models, the first focuses on searching for sequence motifs that are typical of positive peptides on the background of negative ones. The second attempts to predict T-cell epitopes, focusing primarily on neoantigens in the context of tumors.

For the second prediction tool, the authors report improved performance with respect to existing tools.

Since the performance reported in Figure 3e is based on a very small positive sample, it will be interesting to see some more analysis.

In particular:

1. A full precision-recall curve for the new method.
2. Performance of all pMHC prediction tools is notoriously sensitive to the redundancy between the training and testing sets. It will be interesting to see how similar the correctly predicted peptides are to the training sets (positive and negative) and how similar the wrongly predicted peptides are to these two categories. Such comparison will allow us to assess to what extent the prediction is based on the extraction of general biophysical features and to what extent it relies on sequence similarity.

It may also allow to better assess it in comparison to exiting methods.

Reviewer Comments

We thank the reviewers for their very positive view of our manuscript. All reviewer comments have been addressed in the following which has resulted in modifications to the final manuscript. All changes have been highlighted in the manuscript for clarity. Furthermore, in the preparation of the revised manuscript, we have ensured that it adheres to the *Nature Communications* formatting instructions. This has entailed shortening the title and dividing the Results into subheaded sections.

Reviewer #1 (Remarks to the Author):

Jappe et al. examine the temperature stability of a global set of peptide ligands interrogated by mass spectrometry. This is an important proxy for the stability of peptide class I binding, which is related to peptide immunogenicity, critically important for improving cancer immunotherapy.

In general, this is a fine paper, basically acceptable as is.

A few comments for the authors to ponder.

1. C1R cell line is indicated as "low class I" - it is important to inform the readers that this is due to transcription of class I genes and is not related to down regulation of the antigen processing machinery.

This is now addressed in the manuscript (page 4, lines 78-82).

2. Does heating of lysates induce protein aggregation, particularly related to class I complexes? Does recovery of soluble class I fractions significantly drop with temperature, and could this potential bias be related to the bound peptide affinity?

This is an interesting comment. In our study, at incubation temperatures $>70^{\circ}\text{C}$, we observed that the lysate samples became cloudy, and filtering through spin columns was challenging, suggesting that denatured proteins start aggregating and precipitating at these higher temperatures. Aggregation of HLA-A*02:01 complexes has previously been reported at these temperatures (Fuller et al. 2017). We are observing T_m values at temperatures below 70°C which demonstrates that the release of peptide from the HLA heavy chain takes place prior to HLA protein denaturation and aggregation.

3. Since the temperature treatments are done using whole-cell lysates, there are proteases (presumably inhibited, but still...), chaperones, other peptides (even small peptides which might facilitate exchange), and other factors that could impact thermal stability of particular complexes. That's just part of the experimental caveats, which is fine, but would it not have been possible to immunoprecipitate complexes and then subject them to the temperature gradient. Perhaps explain the rationale for choosing your method over others.

We focused on isolation of pHLA complexes that remain after a given thermal treatment since our immunoprecipitation step uses a conformation-specific antibody to capture the remaining, thermally stable, native complexes. Isolating the complexes prior to

treatment would not allow us to make this assessment without a second immunoprecipitation, which would have resulted in unacceptable loss of material.

4. In Figure 3A, is there a reason to plot % rank rather than just the affinities themselves? To better visualize the potential correlation (or lack thereof)?

We agree with the reviewer that it would have been fine to plot the predicted affinities directly; however, we chose to plot the %rank scores in order to bring the affinity distributions of the two alleles (HLA-A*02:01 and HLA-B*07:02) on the same (comparable) scale as binding affinities of allele-specific peptide repertoires vary significantly (Paul et al. 2013).

5. The predictive power could be used to provide a proof of principle of whether thermo-stability leads to longer-lasting complexes, using algorithm-generated examples of equal-affinity peptides that are predicted to have low or high stability. (i.e. made-up peptides that were not identified by MS). Pulse peptides and track via fluorescent label or MS over time. Not required, but would provide excellent proof of principle.

We agree with the reviewer that such a study would provide additional proof of principle to that already provided in our study. We have provided proof of principle instead by carrying out a kinetic stability study of pHLA complexes to calculate the half-lives of the complexes allowing us to determine whether we observe a correlation between kinetic and thermal stability of pHLA. We reasoned that the good correlation observed, which has also been observed previously by others (Blaha et al. 2019), is a validation of higher pHLA thermostability leading to longer-lasting complexes on the cell surface in a physiological milieu.

6. For future studies, consider more physiological exploration of class I peptide complex duration. This could be done by treating cells with BFA for increasing times.

This is very valid input for future studies, as it would allow us to gain further, in-depth insight of the stability of long-lived pHLA on the cell surface.

Reviewer #2 (Remarks to the Author):

The manuscript by Jappe et al examines peptide/HLA complexes by using a novel method that takes advantage of mass spectrometry (MS) to measure, simultaneously, the temperature stability of thousands of peptide complexes. The approach quantitates the amount of specific peptide remaining after incubation of antibody-purified HLA complexes at different temperatures. Previous approaches have shown that the thermal stability of a specific peptide/HLA complex correlates with its potential for immunogenicity. The experimentally determined stability has better predictive ability than current algorithms for binding to HLA. However, these past approaches have required use of recombinant HLA and synthetic peptides to determine thermal stability (e.g. using differential scanning fluorimetry, DSF).

The method by Jappe is very novel, as it not only examines thousands of pep/HLA complexes in a single experiment, but it measures the stability of the complexes isolated

directly from tumor cells. This enables one to assess not just stability, but the abundance of a specific peptide/HLA complex. This is especially relevant to neoantigens that could be used in vaccine formulations as targets for T cell therapies. The study also showed that they could compare relative stabilities of peptides bound to different HLA alleles. Finally, it is significant that they could use their data to develop improved algorithms for predicting pMHC stability.

The only major comment that I have is that I would like to have the authors address the issues of identifying neoantigens among the eluted pep/HLA complexes, and quantitating their stabilities. While they validate their procedure with some neoantigen/HLA complexes that were previously examined by DSF, the question really is what fraction of these would be of sufficient abundance to analyze with their approach. Actual data from a sequenced tumor cell line should be used to address this, but in absence of this they should at the least discuss this issue of abundance of potential neoantigens in the context of their approach.

We agree with the reviewer that the visualisation of neoepitopes will be of great interest, however we feel this is outside the scope of this first paper. The major challenge in this respect is sourcing a tumour cell line in which the immunogenicity of all presented neoantigens has been assessed. Typically only a handful of the predicted neoantigen derived peptides are detected on the surface of tumour cells using untargeted LC-MS/MS approaches (see for instance the work of Bassani-Sternberg 2016). Thus we are working towards a carefully targeted analysis of all possible neoantigen derived peptides in a tumour system (based on whole genome sequencing, identification of simple and complex neoantigens and the prediction of all possible HLA-binding peptides containing these mutations/rearrangements) with matching immunogenicity assessments. This is a major undertaking, which is difficult to complete in the current COVID-19 situation. We hope by publishing our work this will both inspire others and provide sufficient momentum in our project that this type of analysis can be completed in the next 12-18 months, assuming normal access to the laboratory is restored soon. To this end we have altered the discussion to telegraph such experiments.

Minor question, what are the brown bars in Fig 2d?

Fig. 2d shows the distribution of T_m values for peptides restricted by HLA-A*02:01, HLA-B*07:02 and HLA-C*04:01. The bars are indicated with a transparency factor, allowing the distributions to be more obvious for the three alleles; therefore, the brown bars are actually a visual artefact of the overlap of A*02:01 and B*07:02. We have clarified this in the figure text to avoid confusion.

Reviewer #3 (Remarks to the Author):

This study analyzes thermostability of pMHC as a potential new proxy for immunogenicity.

The ideas and suggestions in this study are novel and intriguing and the ms is very well

written.

I was asked to focus my comments on the computational aspects of the ms. In the last part of the study the authors present to predictive models, the first focuses on searching for sequence motifs that are typical of positive peptides on the background of negative ones. The second attempts to predict T-cell epitopes, focusing primarily on neoantigens in the context of tumors.

For the second prediction tool, the authors report improved performance with respect to existing tools.

Since the performance reported in Figure 3e is based on a very small positive sample, it will be interesting to see some more analysis.

In particular:

1. A full precision-recall curve for the new method.

The full precision-recall curves for the prediction of neoepitopes and negative cancer peptides with the Stability Predictor and the other predictors included in the benchmark are shown in Fig. A below. In addition to this, we have included the precision-recall curves for the prediction of neoepitopes and the 199 negative peptides curated from the IEDB with the Stability Predictor and benchmark predictors.

Fig. A. Precision-recall curves demonstrating superior performance of the Stability Predictor. (a) The 26 neoepitopes included in the recent study by Blaha *et al.* (Blaha *et al.* 2019) and the dataset of negative cancer peptides (IEDB) were predicted with the Stability Predictor and the other prediction tools included in this benchmark, which were NetMHCpan-4.0 (Jurtz *et al.* 2017), MixMHCpred (Bassani-Sternberg *et al.* 2017) and MHCFlurry (O'Donnell *et al.* 2018). (b) The same prediction tools as those in a were used to predict the same 26 neoepitopes as well as the 199 confident negative peptides curated from the IEDB.

2. Performance of all pMHC prediction tools is notoriously sensitive to the redundancy between the training and testing sets. It will be interesting to see how similar the correctly predicted peptides are to the training sets (positive and negative) and how similar the wrongly predicted peptides are to these two categories. Such comparison will allow us to assess to what extent the prediction is based on the extraction of general biophysical features and to what extent it relies on sequence similarity. It may also allow to better assess it in comparison to exiting methods.

We appreciate this consideration and have therefore provided visual comparisons of the peptides in the neoepitope/cancer test dataset and the peptides in the positive and negative training data, respectively, based on their similarity and prediction scores (Fig. B). Peptides in the test set have been colour-coded according to whether they are true neoepitopes or negative cancer peptides. We have calculated the similarity of the peptides using the BLOSUM65 matrix as follows

$$\text{sim}(A, B) = \sum_i \text{BL65}(a_i, b_i)$$

where A is the peptide in the test dataset and B is the peptide in the training dataset (Mattsson et al. 2016). For each test peptide, we have calculated the BLOSUM65 similarity score between the peptide and each of the peptides in either the positive or negative training dataset and calculated the mean of these scores to obtain only one score for each test peptide.

We show that the similarity between the test and training datasets is minimal and that the correctly predicted peptides are not more similar to the positive data than the wrongly predicted peptides.

Fig. B. Similarity (BLOSUM65) between 9mer peptides in the test and training data. The neoepitopes and cancer peptides in the test data were predicted with the Stability Predictor and the average sequence similarity of each peptide to (a) the peptides in the positive training data and (b) the peptides in the negative training data was calculated using the BLOSUM65 matrix. The dashed line indicates the threshold (prediction value of 0.5) for the peptides predicted to be positive and negative by the Stability Predictor. The true neoepitopes are indicated in green and the true negative cancer peptides are indicated in black. The peptides correctly predicted positive by the Stability Predictor do not appear to be more similar to the positive training data than the peptides wrongly predicted positive.

References

- Bassani-Sternberg M, *et al.* Direct identification of clinically relevant neoepitopes presented on native human melanoma tissue by mass spectrometry. *Nature communications* **7**, 13404 (2016).
- Bassani-Sternberg, Michal, Chloé Chong, Philippe Guillaume, Marthe Solleder, Hui Song Pak, Philippe O. Gannon, Lana E. Kandalaft, George Coukos, and David Gfeller. 2017. “Deciphering HLA-I Motifs across HLA Peptidomes Improves Neo-Antigen Predictions and Identifies Allosteric Regulating HLA Specificity.” *PLoS Computational Biology* **13**

(8): e1005725. <https://doi.org/10.1371/journal.pcbi.1005725>.

- Blaha, Dylan T, Scott D Anderson, Daniel M Yoakum, Marlies V Hager, Yuanyuan Zha, Thomas F Gajewski, and David M Kranz. 2019. "High-Throughput Stability Screening of Neoantigen/HLA Complexes Improves Immunogenicity Predictions." *Cancer Immunology Research* 7: 50–61. <https://doi.org/10.1158/2326-6066.CIR-18-0395>.
- Fuller, Anna, Aaron Wall, Michael Crowther, Angharad Lloyd, Alexei Zhurov, Andrew Sewell, David Cole, and Konrad Beck. 2017. "Thermal Stability of Heterotrimeric PMHC Proteins as Determined by Circular Dichroism Spectroscopy." *Bio-Protocol* 7 (13). <https://doi.org/10.21769/bioprotoc.2366>.
- Jurtz, Vanessa, Sinu Paul, Massimo Andreatta, Paolo Marcatili, Bjoern Peters, and Morten Nielsen. 2017. "NetMHCpan-4.0: Improved Peptide-MHC Class I Interaction Predictions Integrating Eluted Ligand and Peptide Binding Affinity Data." *The Journal of Immunology* 199: 3360–68. <https://doi.org/10.4049/jimmunol.1700893>.
- Mattsson, Andreas Holm, J. V. Kringelum, C. Garde, and M. Nielsen. 2016. "Improved Pan-Specific Prediction of MHC Class I Peptide Binding Using a Novel Receptor Clustering Data Partitioning Strategy." *Hla* 88 (6): 287–92. <https://doi.org/10.1111/tan.12911>.
- O'Donnell, Timothy J., Alex Rubinsteyn, Maria Bonsack, Angelika B. Riemer, Uri Laserson, and Jeff Hammerbacher. 2018. "MHCflurry: Open-Source Class I MHC Binding Affinity Prediction." *Cell Systems* 7 (1): 129-132.e4. <https://doi.org/10.1016/j.cels.2018.05.014>.
- Paul, Sinu, Daniela Weiskopf, Michael A. Angelo, John Sidney, Bjoern Peters, and Alessandro Sette. 2013. "HLA Class I Alleles Are Associated with Peptide-Binding Repertoires of Different Size, Affinity, and Immunogenicity." *The Journal of Immunology* 191 (12): 5831–39. <https://doi.org/10.4049/jimmunol.1302101>.

REVIEWERS' COMMENTS

Reviewer #3 (Remarks to the Author):

The authors addressed my comments.